# Expression analysis of p16 and TOP2A protein biomarkers in cervical cancer lesions and their correlation with clinico-histopathological characteristics in a referral hospital, Tanzania

Zavuga Zuberi[1,2]*, Alex Mremi[3,4], Jaffu O. Chilongola[4,5], George Semango[1], Elingarami Sauli[1]

1 Department of Global Health and Biomedical Sciences, Nelson Mandela African Institution of Science and Technology, Arusha, Tanzania, 2 Department of Science and Laboratory Technology, Dar es Salaam Institute of Technology, Dar es Salaam, Tanzania, 3 Department of Pathology, Kilimanjaro Christian Medical Centre, Moshi, Tanzania, 4 Kilimanjaro Christian Medical University College, Moshi, Tanzania, 5 Kilimanjaro Clinical Research Institute, Moshi, Tanzania

* zavuga.zuberi@gmail.com

**Data Availability Statement:** All relevant data are within the paper and its Supporting Information files.

## Abstract

### Introduction

Biomarkers yield important information for early diagnosis of cervical cancer. However, they are rarely applied for prognosis of cervical cancer in Tanzania, where visual inspection assay with acetic acid or Lugol's iodine and Pap test are being used as the standard screening/ diagnostic methods.

### Methods

This was a retrospective hospital-based cross-sectional study that was conducted to assess cyclin-dependent kinase inhibitor (p16) and topoisomerase II-alpha (TOP2A) proteins expression among women seeking cervical cancer care at Kilimanjaro Christian Medical Centre, Tanzania between May 1, 2017 and May 10, 2018. Immunohistochemistry technique was used to detect the expressions of p16 and TOP2A proteins from the retrieved formalin-fixed and paraffin-embedded (FFPE) cervical biopsies.

### Results

A total of 145 patients, with a mean age of 52.1 ± 12.9 years, were included in this study. Upon immunohistochemistry staining, 103 (71.0%) and 90 (62.1%) were p16 and TOP2A positive respectively. There was a strong association between histopathological class and p16/TOP2A expression levels (Fisher's exact test, $p<0.001$). Moreover, there was a strong positive correlation between p16/TOP2A and cancerous cervical lesions (Spearman's rank correlation coefficients = 0.833 and 0.687, $p = 0.006$ and 0.005, respectively). The age-adjusted odds ratio for predicting cervical cancer lesions were independently significant for p16/TOP2A biomarkers in FFPE cervical tissues [p16: OR = 1.142 (95% CI: 1.059–1.232, $p<0.001$) and TOP2A: OR = 1.046 (95% CI: 1.008–1.085, $p = 0.015$)]. Importantly, the

**Funding:** This study received financial support from the World Bank under the CREATES-FNS project at NM-AIST (grant no. 02090107-048-301-4001-P044-J01S01-C42). Funder's URL: https://www.creates-nmaist.ac.tz/ The funders had no role in study design, data collection and analysis, decision to publish, or preparation of the manuscript.

**Competing interests:** The authors have declared that no competing interests exist.

diagnostic performance of p16 was higher than that of TOP2A in the diagnosis of cancerous lesions from non-cancerous cervical lesions (sensitivity: 97.2% *versus* 77.6%, accuracy: 92.8% *versus* 87.8%, respectively).

## Conclusion

Our study has highlighted that over-expression of TOP2A is related to the grade of cervical intraepithelial neoplasia but does not predict prognosis in cervical cancer. Similarly, expression of p16 is related to degree of histological dysplasia and malignancy, suggesting its prognostic and predictive value in the management of cervical cancers. Further bigger studies are needed to validate their applications in the early diagnosis of cervical cancer.

## Introduction

Cervical cancer is the fourth common cancer affecting women of reproductive ages globally [1, 2]. According to the data from GLOBOCAN, it was estimated that about 18.1 million new cancer cases and 9.6 million cancer deaths occurred worldwide in 2018 [1]. Among these, 570,000 were new cervical cancer cases, and 311,000 were cervical cancer deaths, accounting for 6.6% and 7.5% of all new cancer cases, and cancer-related deaths among women, respectively [1]. Many high-income countries have successfully reduced cervical cancer incidence and mortality, as they have effective population-based early detection and screening programs that are well organized and with high coverage rates. However, cervical cancer early diagnosis programs have not been able to control the disease in low-resource countries like Tanzania, partly because of lack, or limited access and low performances of available methods [1, 2].

In Tanzania, cervical cancer accounts for 39.0% of new cancer cases and 40.6% of deaths among all women cancers, a higher rate than the average global cancer burden [1]. These data rank cervical cancer as the pre-eminent cause of cancer-related deaths in women of all ages in Tanzania [3]. This is contributed by the high rate of human papillomavirus (HPV) infection, advanced stage of cervical cancer when diagnosed, a limited number of effective and sustainable diagnostic programs, inadequate cervical cancer screening awareness campaigns, inaccessibility of diagnostic and treatment services, treatment abandonment, excess relapse, etc. [4, 5].

Importantly, combined efforts including knowledge creation and awareness campaigns, are highly needed to improve cervical cancer screening among women in rural Tanzania [6]. Yet, poor response to cervical cancer early diagnosis has been reported to be significantly influenced by distant screening centers, and limited knowledge of cervical cancer among Tanzanian women [7, 8]. Moreover, immunohistopathological assays require well-educated and qualified staffs as well as high-quality control in test implementation, that, unfortunately is not provided in Tanzania [4, 9, 10]. The most used cervical cancer diagnosis methods in Tanzania are visual inspection-based methods with acetic acid or Lugol's iodine, while the conventional cytology (Pap test) is not frequently used [4, 11].

Currently, the national cervical cancer screening programme in Tanzania uses visual inspection with acetic acid (VIA) as the standard screening procedure [11, 12] which is available free of charge in some governmental- and faith-based hospitals on different levels of care. Although, VIA has lower sensitivity and specificity compared to Pap test and HPV DNA testing [4, 11], it remains the standard of care in many low-income countries because of its single visit approach and generally high prevalence of cervical cancer in these countries. However, Pap test is available in the zonal hospitals in Tanzania.

In Tanzania, very limited studies estimating the coverage of cervical cancer screening by age exist [12]. However, a follow-up study that enrolled women aged between 25 and 60 years from three cervical cancer screening clinics from urban and semi-rural areas in Tanzania reported 17.2% (696/4,043) and 14.2% (438/3,074) of the women aged between 25 and 60 years had high-risk HPV (hr-HPV) at baseline and in the first follow-up, respectively [13]. Only 3.4% (139/4,043) of the women aged between 25 and 60 years had high-grade squamous intraepithelial lesion (HSIL) in the same study [13]. Another study by Dartell et al. [11] reported lower sensitivity of VIA than HPV-testing in detecting HSIL cervical lesions during a hospital-based cervical cancer screening event in Dar es Salaam, Tanzania. In many high-income countries, cervical cancer rates declined substantially after the wide spread introduction of the Pap test in the mid 20[th] century [12]. However, similar trends did not emerge in low- and middle-income countries including Tanzania due to lack of resources for screening implementation including cytology review and low population coverage leading to advanced detection of cervical cancer and poor survival rates [2, 12].

Several histological studies have reported the application of most common cellular biomarkers, including Ki-67, cyclin-dependent kinase inhibitor (p16), and ProEx C [topoisomerase II-alpha (TOP2A) and minichromosomal maintenance-2 (MCM2)] in the early diagnosis of cervical cancer [14–17]. Recently, Ki-67 has been demonstrated to be an important cell proliferation biomarker for malignant lesions among women in Tanzania [18, 19]. However, several cancer biomarkers, especially when applied individually, may not detect cervical cancer as it develops, making them unsuitable for early diagnosis of cervical cancer [16, 20, 21]. However, a combination of biomarkers may improve the sensitivity and specificity of cancer detection in discriminating altered cells from normal cells across age groups [14, 15, 21–23]. During the pathogenesis of cervical cancer, the expression of E7 viral oncoprotein inactivates retinoblastoma proteins (pRb), which consequently increases the E2F activation factors in the cell, which leads to increased p16 in the S-phase of the cell cycle. Likewise, the TOP2A gene encodes for DNA topoisomerase, a nucleic enzyme responsible for unwinding supercoiled DNA strands during its replication in the S-phase of the cell cycle [22, 24]. Expression levels of p16 and TOP2A biomarkers have been reported for early diagnosis of cervical cancer in high-resource countries, [14, 15, 22, 25] which necessitates the need for their feasibility study in low-resource settings.

These cellular biomarkers (p16 and TOP2A) are emerging as novel biomarkers for early diagnosis and prognosis of cervical cancer [16, 20, 26]. However, very limited studies have been conducted to investigate their application as early diagnostic markers in improving the accuracy of clinico-histopathological diagnosis of cervical lesions in resources limited settings with a high burden of cervical cancer [4, 9, 10, 23, 24]. This may contribute to insufficient data being reported on the prevalence of cervical cancer in Tanzania. Likewise, low-cost, sensitive and specific screening methods including low-cost HPV DNA tests such as *care*HPV for detection of hr-HPV genotypes are highly needed in Tanzania [27]. Therefore, this study was conceived to evaluate the usefulness of p16 and TOP2A as potential biomarkers in dysplastic and malignant alteration of cervical epithelium by analyzing a series of benign, precancerous and cancerous cervical lesions so as to assess whether their expression might be of any use in predicting prognosis in cervical carcinogenesis in Tanzania.

## Materials and methods

### Retrieval of study samples and related clinical information

This was a retrospective hospital-based cross-sectional study that was conducted using cervical biopsy samples that were taken from women seeking care at Kilimanjaro Christian Medical

Centre (KCMC) from May 1, 2017 to May 10, 2018. KCMC is a referral, consultant research, and teaching hospital located in Kilimanjaro, Northern Tanzania. It provides medical services to approximately 15.7 million people from Kilimanjaro, Tanga, Manyara, and Arusha regions with more than 4,500 histology/cytology samples processed and examined per year.

The inclusion criteria involved patients with complete clinico-histopathological information and good morphology of their cervical biopsies. The exclusion criteria involved missing tissue blocks and incomplete clinical histopathological information. This study also included all formalin-fixed and paraffin-embedded cervical biopsy stained by routine haematoxylin and eosin (H&E) staining technique at KCMC between May 1, 2017 and May 10, 2018. H&E-stained slides were retrieved and reviewed by two qualified independent histopathologists.

The histopathological classification of cervical cancer cases was performed according to the World Health Organization classification of tumors from women reproductive organs [28]. In addition, epithelial cells were classified into how much the epithelial cells were affected into a sub-class 'precancerous cervical lesion' having three-tiered categories: cervical intraepithelial neoplasia (CIN)-1), CIN-2 and CIN-3. Moreover, a sub-class 'benign cervical lesion' was classified into cervicitis, endocervical polyps, and others including nabothian cyst, cervical koilocytosis, and cervical papilloma.

## Immunohistochemistry (IHC) assays

From the selected tissue blocks, two slides (labelled as p16 and TOP2A) were sectioned at 3μm and placed on the DFrost Plus positively charged slides (Diapath S.P.A., Martinengo BG, Italy) and baked in a hot air oven at 40˚C overnight. Slides with tissue sections were then deparaffinized using two changes of xylene for 8 min each. The samples were thereafter rehydrated using decreasing grades of ethanol solution (100%, 95%, 80%, and 70%, respectively), with 10 dips in each solution.

All these procedures were performed in the humidity chamber to avoid drying up of slides between the steps. During IHC staining, the tissue sections were circled with a hydrophobic pen (Dako Denmark A/S, Glostrup, Denmark) and endogenous peroxide activity was blocked using a ready-to-use peroxidase-blocking reagent (Dako Denmark A/S, Glostrup, Denmark) for 15 min. After rinsing with distilled water for 3 min, antigen retrieval was performed by pre-heating at 65˚C for 10 min into a pressure cooker (Kanchan International Ltd., Mumbai, India). Microscope slides were then heated in a cold citrate buffer retrieval solution at pH 6.0. Slides were cooled using tap water for another 10 min, followed then by rinsing with 20x diluted wash buffer for 5 min.

All sections were incubated with pre-diluted primary antibody p16 clone G175-405 (Medaysis Company, Livermore, CA, USA) and TOP2A clone Ki-S1 pre-diluted primary antibody (Medaysis Company, Livermore, CA, USA), respectively for 30 min. The sections were then washed with 20x wash buffer for about 5 min, followed by incubation with universal horseradish peroxidase (HRP) for 30 min and washing with phosphate-buffered saline (PBS) twice each for 3 min. Moreover, sections were then incubated with 3, 3' diaminobenzidine DAB + Chromogen (Dako Denmark A/S, Glostrup, Denmark) for 10 min and then rinsed with water for 2 min.

The sections were thereafter counterstained with hematoxylin for 17 dips and bluing for 5 min. Sections were then dehydrated in the ascending grades of ethanol solution (70%, 80%, 95%, and 100%, respectively), and then cleared in two changes of xylene for 5 min each. Finally, the sections were covered using mounting medium by using Tissue-Tek® Coverslipper (Sakura Finetek Inc., Torrance, CA, USA). In reducing false-positive IHC results, two positive controls were used; for p16, a cervical squamous cell carcinoma tissue was used, and for TOP2A, a breast carcinoma tissue was used.

## Immunohistochemistry (IHC) assessment

The tissue sections were microscopically evaluated by counting the number of proliferating cells and intensity of stained cells for p16 and TOP2A. Positive cells for p16 were given an immunoscore of 1–2+ to indicate high cell proliferation or high intensity, while an immunoscore of 0 was given to indicate the absence of cell proliferation or low intensity [29]. For positive TOP2A cells, an immunoscore of 1–2+ was given to indicate moderate cell proliferation or moderate intensity, an immunoscore of 3+ was given to indicate high cell proliferation or high intensity while an immunoscore of 0 indicated the absence of cell proliferation or low intensity [30].

## Statistical analyses

Clinico-histopathological information and biomarkers expression data were entered into the Microsoft® Excel 2013 and exported to R software version 4.0.2 (https://www.r-project.org) for statistical analyses as presented in tables and figures. Descriptive statistics, such as mean, frequency, and standard deviation were used to summarize numerical data. The associations between clinico-histopathological information and biomarkers expression were performed using Fisher's exact test. Moreover, a logistic regression model was used to estimate relationships between the expression of p16 and TOP2A biomarkers with the development of cervical cancer lesions by estimating age-adjusted odds ratio (OR) with 95% confidence intervals (CIs). In addition, the performance of p16 and TOP2A biomarkers was evaluated by assessing their sensitivity, specificity, and accuracy in differentiating cancerous cervical lesions from benign and precancerous cervical lesions. Furthermore, Spearman's rank correlation test was used to assess the strength of relationships between age, histopathological factors, and biomarkers expression. The results with two-sided $p<0.05$ were considered statistically significant.

## Ethical clearance and approval

Ethical clearance was obtained from the Medical Research Coordinating Committee (MRCC) of the Tanzania National Institute for Medical Research (NIMR), with permit number NIMR/HQ/R.8a/Vol. IX/2764. A formal letter was written to obtain permission from the Pathology Department of the Kilimanjaro Christian Medical Center. Written informed consent was not applicable due to the retrospective design of this study.

# Results

## Clinico-histopathological features and biomarkers expression among women seeking cervical cancer care

A total of 145 cervical biopsies and patient's information retrieved from the Pathology department archival material between May 1, 2017 and May 10, 2018 were included in this study. All the cervical biopsies were used for immunohistochemistry staining of p16 and TOP2A biomarkers. The included cervical biopsies were considered based on clinical histopathological information and good morphology.

Patients' age ranged from 23 to 83 years, with mean and standard deviation of 52.1±12.9 years respectively. Of the 145 retrieved cervical biopsies, 6 (4.1%) had non-definitive diagnosis by histology, 95 (65.4%) were cervical cancer, while 33 (22.7%) and 11 (7.6%) had benign and precancerous cervical lesions, respectively (Table 1).

Histopathological analysis showed that, squamous cell carcinoma accounted for 83 (57.2%) of the total cervical biopsies, followed by adenocarcinoma, which accounted for 6 (4.1%),

**Table 1. Clinico-histopathological features among women seeking cervical cancer care at Kilimanjaro Christian Medical Centre (n = 145).**

| Characteristics | N = 145 |
|---|---|
| | n (%) |
| **Age (years)** | 52.1 ± 12.9 [†] |
| **Histopathological subtype** | |
| Benign cervical lesion | |
| Cervicitis | 15 (10.3) |
| Endocervical polyps | 15 (10.3) |
| Others | 3 (2.1) |
| **Sub-total** | 33 (22.7) |
| Precancerous cervical lesion | |
| CIN-1 | 7 (4.8) |
| CIN-2 | 1 (0.7) |
| CIN-3 | 3 (2.1) |
| **Sub-total** | 11 (7.6) |
| Cancerous cervical lesion | |
| Squamous cell carcinoma | 83 (57.2) |
| Adenocarcinoma | 6 (4.1) |
| Others | 6 (4.1) |
| **Sub-total** | 95 (65.4) |
| Non-definitive diagnosis | 6 (4.1) |
| **Total** | **145 (100)** |
| **Clinical symptoms** | |
| PV bleeding | 62 (42.7) |
| Vaginal discharge | 42 (29.0) |
| Lower abdominal pain | 34 (23.4) |
| Others | 7 (4.8) |
| **Biomarkers expression** | |
| p16 | |
| Low | 42 (29.0) |
| High | 103 (71.0) |
| TOP2A | |
| Low | 55 (37.9) |
| Moderate | 29 (20.0) |
| High | 61 (42.1) |

[†] mean ± standard deviation

CIN cervical intraepithelial lesion

LSIL low-grade squamous intraepithelial lesion

HSIL high-grade squamous intraepithelial lesion

whilst others were undifferentiated carcinoma and neuroendocrine carcinoma as summarized in Table 2. Both cervicitis and endocervical polyps were dominant benign conditions, of which each account for 15 (10.3%) of all cervical lesions. On the other hand, the cervical low-grade squamous intraepithelial lesion (LSIL) was exhibited in 7 (4.8%) of all cervical lesions.

Furthermore, immunohistochemistry staining (Fig 1) revealed that, among 145 patients seeking cervical cancer care, 103 (71.0%) and 61 (42.1%) strongly expressed p16 (p16+) and TOP2A (TOP2A++) proteins, respectively (Table 1). Importantly, the expressions of p16+ and TOP2A++ proteins were predominantly found among women aged 50–59 years (Fig 2).

**Table 2. Association between clinico-histopathological features and protein expressions among women seeking care at Kilimanjaro Christian Medical Centre (n = 145).**

| Variables | N = 145 | p16 protein expression | | p-value[a] | TOP2A protein expression | | | p-value[a] |
|---|---|---|---|---|---|---|---|---|
| | | Low | High | | Low | Moderate | High | |
| | n (%) | n (%) | n (%) | | n (%) | n (%) | n (%) | |
| **Age (years)** | | | | 0.813 [b] | | | | 0.256 [b] |
| <30 | 5 (3.4) | 2 (40.0) | 3 (60.0) | | 3 (60.0) | 1 (20.0) | 1 (20.0) | |
| 30–39 | 21 (14.5) | 7 (33.3) | 14 (66.7) | | 11 (52.4) | 3 (14.2) | 7 (33.3) | |
| 40–49 | 48 (33.1) | 16 (33.3) | 32 (66.7) | | 20 (41.7) | 12 (25.0) | 16 (33.3) | |
| 50–59 | 34 (23.4) | 8 (23.5) | 26 (76.5) | | 11 (32.4) | 7 (20.6) | 16 (47.1) | |
| 60–69 | 25 (17.2) | 7 (28.0) | 18 (72.0) | | 7 (28.0) | 2 (8.0) | 16 (64.0) | |
| >70 | 12 (8.3) | 2 (16.7) | 10 (83.3) | | 3 (25.0) | 4 (33.3) | 5 (41.7) | |
| **Histopathological class** | | | | <0.001 | | | | <0.001 |
| Benign | 33 (22.8) | 33 (100) | 0 (0) | | 30 (90.9) | 2 (6.1) | 1 (3.0) | |
| Precancerous | 11 (7.6) | 2 (18.2) | 9 (81.8) | | 8 (72.7) | 2 (18.2) | 1 (9.1) | |
| Cancerous | 95 (65.5) | 1 (1.1) | 94 (98.9) | | 11 (11.6) | 25 (26.3) | 59 (62.1) | |
| Non-definitive diagnosis | 6 (4.1) | 6 (100) | 0 (0) | | 6 (100) | 0 (0) | 0 (0) | |
| **Histopathological subtype** | | | | | | | | |
| **Benign cervical lesion** | | | | 1.000 | | | | 0.479 [b] |
| Cervicitis | 15 (10.3) | 15 (100) | 0 (0) | | 13 (86.7) | 1 (6.7) | 1 (6.7) | |
| Endocervical polyps | 15 (10.3) | 15 (100) | 0 (0) | | 14 (93.3) | 1 (6.7) | 0 (0) | |
| Others | 3 (2.1) | 3 (100) | 0 (0) | | 2 (66.7) | 1 (33.3) | 0 (0) | |
| **Precancerous cervical lesion** | | | | 1.000 [b] | | | | 0.241 [b] |
| CIN-1 | 7 (4.8) | 2 (28.6) | 5 (71.4) | | 6 (85.7) | 1 (14.3) | 0 (0) | |
| CIN-2 | 1 (0.7) | 0 (0) | 1 (100) | | 0 (0) | 0 (0) | 1 (100) | |
| CIN-3 | 3 (2.1) | 0 (0) | 3 (100) | | 2 (66.7) | 1 (33.3) | 0 (0) | |
| **Cancerous cervical lesion** | | | | 0.122 [b] | | | | 0.584 [b] |
| Squamous cell carcinoma | 83 (57.2) | 0 (0) | 83 (100) | | 10 (12.0) | 23 (27.7) | 50 (60.2) | |
| Adenocarcinoma | 6 (4.1) | 1 (16.7) | 5 (83.3) | | 1 (16.7) | 0 (0) | 5 (83.3) | |
| Others | 6 (4.1) | 0 (0) | 6 (100) | | 0 (0) | 2 (33.3) | 4 (66.7) | |
| **Non-definitive diagnosis** | 6 (4.1) | 6 (100) | 0 (0) | | 6 (100) | 0 (0) | 0 (0) | |

[a]Fisher's exact test

[b] simulated p-value

CIN cervical intraepithelial lesion

## Association between clinico-histopathological features and biomarkers expression

Age categories were not significantly associated with expression levels of p16 and TOP2A among women seeking cervical cancer care. In contrary, there was significant association between histopathological class and expression levels of p16 and TOP2A among women seeking cervical cancer care ($p<0.001$) (Table 2). These biomarkers may therefore be potentially used in classifying cervical cancer lesions. However, no significant associations were observed between the histopathological subtypes of benign, precancerous, and cancerous cervical lesions, with p16 and TOP2A expressions in the studied population (Table 2). This implies p16 and TOP2A cannot be potentially used as prognostic biomarkers across all age groups.

Moreover, there was significant positive correlation between p16 expression and cancerous cervical lesions in the studied population (Spearman's rank correlation

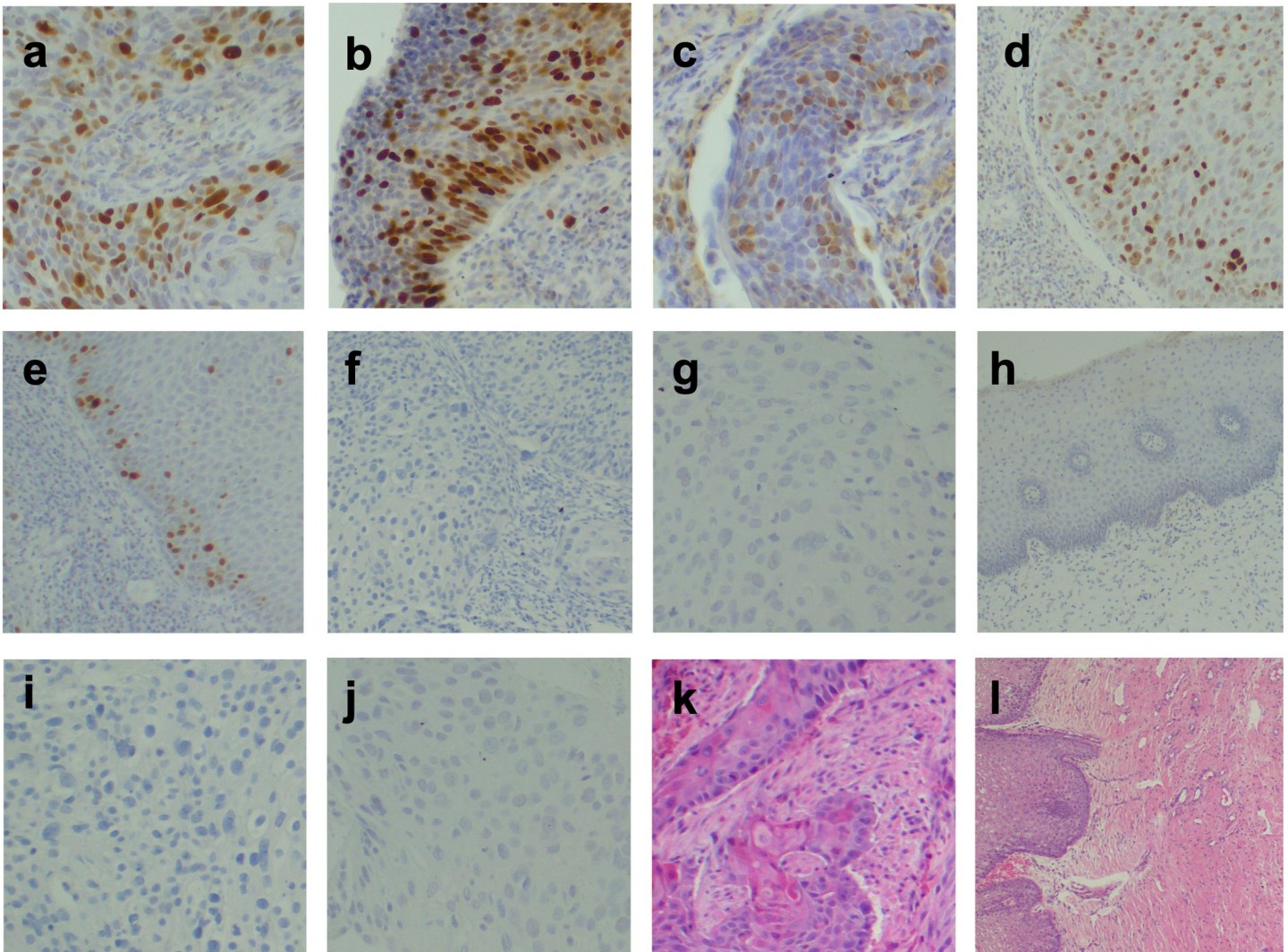

**Fig 1. Monographs.** Nucleus and cytoplasm were positively stained for (**a-b**) p16 at 200x for cancerous and precancerous cervical lesion respectively. Nucleus positively stained for (**c-d**) TOP2A at 200x for cancerous and precancerous cervical lesion respectively, and (**e**) TOP2A at 40x for a benign cervical lesion. Nucleus and cytoplasm negatively stained for (**f-g**) p16 at 100x and 200x for cancerous and precancerous cervical lesion respectively, and (**h**) p16 at 200x for a benign cervical lesion. Nucleus negatively stained for (**i-j**) TOP2A at 200x for cancerous and precancerous cervical lesion respectively; (**k-l**) H&E staining at 100x and 40x for squamous cell carcinoma and normal cervix respectively.

coefficient = 0.833, $p$ = 0.006). In contrary, a non-significant positive correlation between age and p16 expression was observed (Spearman's rank correlation coefficient = 0.110, $p$ = 0.643). By considering the effect size rather than $p$-value due to small sample size, there was a difference in the relationship between benign cervical lesions and p16 expression (Spearman's rank correlation coefficient = -0.944, $p<0.001$) (Table 3). A significant positive correlation was observed between TOP2A expression and cancerous cervical lesion in the studied population (Spearman's rank correlation coefficient = 0.687, $p$ = 0.005). Moreover, there was a significant positive correlation between p16 and TOP2A expressions (Spearman's rank correlation coefficient = 0.605, $p$ = 0.020) (Table 3). Therefore, the cancerous cervical lesions strongly correlated with p16 and TOP2A expressions in the studied population. The expression of p16 also moderately correlated with TOP2A expression.

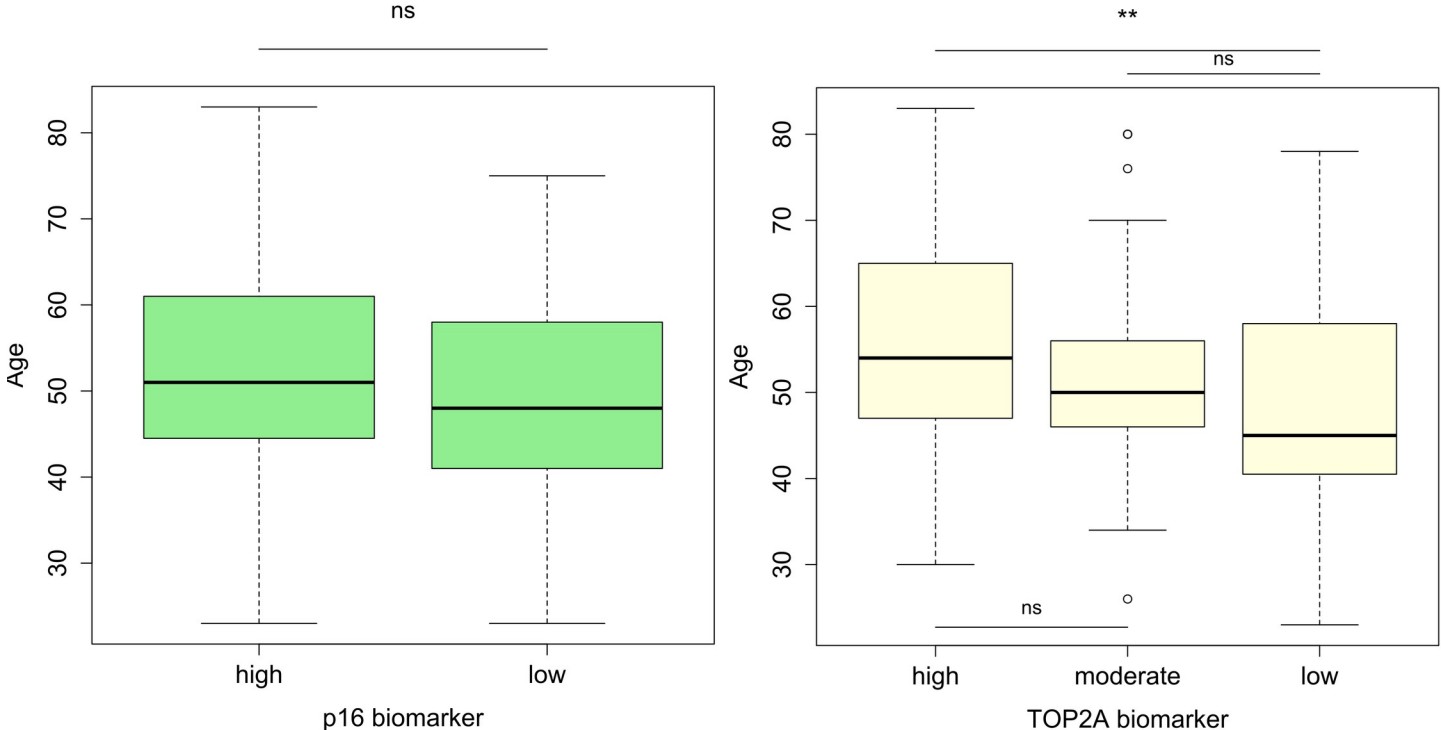

**Fig 2. Distribution of p16 and TOP2A biomarkers expression across age-groups among women seeking care at Kilimanjaro Christian Medical Centre (n = 145).**
The bold horizontal line within each column represents medians. Statistical significance is marked with double asterisks (Student's *t*-test, *p* = 0.006) and *ns* stands for not significant.

### Diagnostic performance and strength of associations between p16 and TOP2A immunohistochemistry in differentiating cancerous cervical lesions from benign and precancerous cervical lesions

The diagnostic utility of p16 and TOP2A immunohistochemistry was evaluated by classifying cancerous cervical lesions as positive cases, with benign and precancerous cervical lesions altogether classified as negative cases. The diagnostic performance of p16 was higher than that of TOP2A in the diagnosis of cancerous lesions from non-cancerous cervical lesions (sensitivity:

**Table 3. Correlation matrix for p16 and TOP2A biomarkers expression with the histopathological factors among women seeking cervical cancer care at Kilimanjaro Christian Medical Centre (n = 139).**

|  | Age | Benign | Precancerous | Cancerous | TOP2A | p16 |
|---|---|---|---|---|---|---|
| **Age** | 1.000 |  |  |  |  |  |
| **Benign** | -0.144 | 1.000 |  |  |  |  |
| **Precancerous** | -0.253 | 0.164 | 1.000 |  |  |  |
| **Cancerous** | 0.278 | -0.820* | -0.431 | 1.000 |  |  |
| **TOP2A** | 0.240 | -0.600* | -0.238 | 0.687** | 1.000 |  |
| **p16** | 0.110 | -0.944*** | 0.052 | 0.833** | 0.605* | 1.000 |

*p*< 0.05 was considered statistically significant.

* indicates significant (*p*< 0.05)

** indicates highly significant (*p* < 0.01)

*** indicates very highly significant (*p*< 0.001)

97.2% *versus* 77.6%, accuracy: 92.8% *versus* 87.8%). In contrary, a combination of p16 and TOP2A immunohistochemistry for the diagnostic utility was slightly lower than that of p16 or TOP2A alone in differentiating cancerous cervical lesions from non-cancerous cervical lesions (Table 4). These findings imply that p16 may be a potential biomarker in differentiating cancerous cervical lesions from benign and precancerous cervical lesions in the studied population.

Nevertheless, cancerous cervical lesions were statistically correlated with p16 expression ($p<0.001$). Likewise, cancerous cervical lesions were significantly correlated with expression of TOP2A ($p = 0.014$). In contrary, the combined expression of p16 and TOP2A biomarkers was non-significantly correlated with cancerous cervical lesions in the studied population (Table 5). Moreover, the age-adjusted odds ratio for the cancerous cervical lesions (relative to the benign and precancerous cervical lesions) were 1.142 (95% CI: 1.059–1.232) and 1.046 (95% CI: 1.008–1.085) for the expression levels of p16 and TOP2A biomarkers, respectively. In addition, the age-adjusted odds ratio for the cancerous cervical lesions (relative to the benign and precancerous cervical lesions) was 0.989 (95% CI: 0.946–1.034) for the combination of p16 and TOP2A biomarkers (Table 5). Overall, our findings support the associations of p16 and TOP2A biomarkers in the development of cervical cancer lesions.

## Discussion

Our study findings showed that the majority of women had overexpression of p16 and TOP2A biomarkers. These study findings correspond with previously reported findings which suggested that p16 and TOP2A expression increases with severity of cervical lesions, and may thus be used as markers in the early diagnosis of cervical lesions [14, 22, 23, 31]. In the present study, significant associations were observed between histopathological class with expression levels of p16 and TOP2A biomarkers (each with $p<0.001$).

Expressions of p16 and TOP2A were frequently observed in the 40–49 years age group in this current study, which is considered to be the high-risk age group for precancerous and cancerous cervical lesions. This can be explained by the limited cervical cancer early diagnosis practices and lack of awareness campaigns for Tanzanian women in preventing the occurrence of cervical cancer [6–10]. These challenges could be responsible for increased incidence and mortality rates of cervical cancer in older women [8, 32]. Moreover, introduction of immunohistochemistry assessment of biomarkers may aid in the histopathological classification of cervical cancer and other cancers, and may also assist in early identification of Tanzanian women who are at high risk for recurrence of cervical cancer.

Correlating p16 expression with clinico-histopathological factors revealed that p16 expression was strongly positively correlated with cancerous cervical lesions, where 100% of

**Table 4. Diagnostic values of p16 and TOP2A immunohistochemistry in differentiating cancerous cervical lesions from benign and precancerous cervical lesions among women seeking cervical cancer care at Kilimanjaro Christian Medical Centre (n = 139).**

| Methods | p16 | TOP2A[a] | p16 *plus* TOP2A[a] |
|---|---|---|---|
| | % (95% CI) | % (95% CI) | % (95% CI) |
| Sensitivity | 97.2 (85.5–99.9) | 77.6 (63.4–88.2) | 85.9 (76.6–92.5) |
| Specificity | 91.3 (84.1–95.9) | 93.3 (86.1–97.5) | 92.2 (87.5–95.6) |
| Accuracy | 92.8 (87.2–96.5) | 87.8 (81.1–92.7) | 90.3 (86.2–93.5) |

CI confidence intervals

[a] modified expression levels (low: moderate + high)

**Table 5. Multivariate regression analysis and the strength of association between expression levels of p16 and TOP2A in differentiating cancerous cervical lesions from benign and precancerous cervical lesions among women seeking cervical cancer care at Kilimanjaro Christian Medical Centre (n = 139).**

| Variable [a] | Factor | Overall model significance $p<0.001$ | | | |
|---|---|---|---|---|---|
| | | AOR | 95% CI for AOR | | *p*-value |
| p16 | Low | 1 | | | |
| | High | 1.142 | 1.059 | 1.232 | **<0.001** |
| TOP2A | Low | 1 | | | |
| | High | 1.046 | 1.008 | 1.085 | **0.015** |
| p16 *and* TOP2A | Low | 1 | | | |
| | High | 0.989 | 0.946 | 1.034 | 0.638 |

CI confidence intervals

AOR age-adjusted odds ratio

[a] benign and precancerous cervical lesions were altogether used as a reference group

squamous cell carcinoma and 83.3% adenocarcinoma had overexpression of p16, which was also in accordance with previously reported studies [33, 34]. In addition, another meta-analysis study reported the prognostic significance of p16 overexpression with improved overall survival and disease-free survival in cervical cancer patients [35]. Strong correlation between p16 expression level and cancerous cervical lesions can be explained by overexpression of p16 due to pRb inactivation in the cancerous cervical lesions than non-cancerous cervical lesions [25]. However, stratification of cancerous cervical lesions to associate the p16 expression and HPV subtype was not performed in our study.

Furthermore, there was also significant positive correlation between TOP2A expression levels and cancerous cervical lesions. TOP2A overexpression was observed in 60.2% of squamous cell carcinoma and 83.3% of adenocarcinoma. Our findings agree with those reported by Del Moral-Hernández et al. [24], which showed that increased expression of TOP2A/MCM2 biomarkers to approximately 3-times increased progression risk of HSIL to cervical cancer lesions. Similarly, TOP2A expression levels discriminated dysplastic and non-dysplastic FFPE cervical tissues with improved detection of CIN2+ [15, 36]. In addition, a 40% difference in the expression of p16 (high) versus TOP2A (high) biomarkers (100% versus 60%) observed in the squamous cell carcinoma may imply that the p16 biomarker is more specific for squamous cell carcinoma than the TOP2A biomarker [22, 24, 31, 37]. Furthermore, immunoscoring of the cut-off threshold for positive cells for both p16 and TOP2A biomarkers might vary between pathologists resulting in significant differences.

In a multivariate logistic regression model, the age-adjusted OR for predicting cervical cancer lesions were independently significant for p16/TOP2A biomarkers in archived FFPE cervical tissues [p16: OR = 1.142 (95% CI: 1.059–1.232, $p<0.001$) and TOP2A: OR = 1.046 (95% CI: 1.008–1.085, $p = 0.015$)]. These findings corroborate with previously reported studies that used cervical scraps/ FFPE tissues, which showed that progression of LSIL/CIN-1 to cervical cancer lesions was conferred by p16, TOP2A, p16 and TOP2A [ProEx C] [24, 38, 39]. However, our results herein disagree with those reported by Peres et al. [22], that high TOP2A expression levels were observed in cervical smear samples than in cervical biopsies. Furthermore, combination of p16 and TOP2A biomarkers revealed non-significant correlation, which was not useful in classifying cervical lesions, but this needs to be confirmed by studies with a large number of cases [21, 23, 25, 31].

p16 is a tumour suppressor protein that accumulates in the nucleus and cytoplasm of cervical cells transformed by hr-HPV [24, 29, 31]. In this study, the sensitivity and specificity of p16

were greater than 90% in discriminating cancerous cervical lesions from non-cancerous cervical lesions, suggesting it to be a possibly very useful marker for early diagnosis when discriminating between healthy and diseased women seeking cervical cancer care at KCMC, which is also in agreement with findings reported by Espinosa et al. [40]. Similarly, TOP2A is a nuclear protein that controls DNA replication and chromosome separation of aberrant cervical cells in the S-phase of the cell cycle [22, 24]. Nevertheless, a 17% decreased sensitivity of TOP2A relative to p16 was observed in this study. This implies that 20% of diseased women (false-negatives) will possibly be misdiagnosed. Our findings are in agreement with reported sensitivity and specificity for TOP2A mRNA biomarker as reported in previous studies [41, 42]. Generally, p16 was demonstrated to confer highly accurate diagnostic performance than TOP2A biomarker in effectively differentiating cancerous and non-cancerous cervical lesions in the studied population.

Limitations of this study include a relatively low number of analyzed archived cervical biopsies which could be contributed by missing of the patients' data based on the retrospective study design. Due to limited resources, genotyping of HPV subtypes was not performed which could predict the risks of cervical cancer lesions and its precursors development in association with the expression of p16 and TOP2A biomarkers in the studied population, and Tanzanian population at large.

In conclusion, over-expression of p16 and TOP2A proteins significantly correlated with cancerous cervical lesions, and may be promising biomarkers for discriminating cancerous cervical lesions from non-cancerous cervical lesions in the studied population. However, further investigation and feasibility studies are still needed on these markers before considering them as biomarkers for early diagnosis of cervical cancer especially in Tanzania regions that have high-burden of cervical cancer cases. In addition, the diagnostic performances of these biomarkers need to be validated in clinical setting and compared to readily available commercial diagnostic assays for cervical cancer. Importantly, a large study involving a combination of p16 and TOP2A proteins and their mRNAs, and their association with hr-HPV subtypes can provide a more detailed pattern in the study area.

## Supporting information

**S1 Table. Deidentified raw dataset for the study participants.**
(XLS)

## Acknowledgments

We greatly acknowledge Mr. Erick P. Magorosa for the Histotechnology experimental analyses, all colleagues from the pathology department at KCMC.

## Author Contributions

**Conceptualization:** Zavuga Zuberi, Alex Mremi, Elingarami Sauli.

**Data curation:** Zavuga Zuberi, Alex Mremi.

**Formal analysis:** Zavuga Zuberi, Alex Mremi.

**Investigation:** Zavuga Zuberi, Alex Mremi, George Semango, Elingarami Sauli.

**Methodology:** Zavuga Zuberi, Alex Mremi, George Semango, Elingarami Sauli.

**Project administration:** Alex Mremi, Elingarami Sauli.

**Resources:** Alex Mremi, Jaffu O. Chilongola, Elingarami Sauli.

**Validation:** Jaffu O. Chilongola.

**Visualization:** Alex Mremi, Elingarami Sauli.

**Writing – original draft:** Zavuga Zuberi, Alex Mremi.

**Writing – review & editing:** Zavuga Zuberi, Alex Mremi, Jaffu O. Chilongola, George Semango, Elingarami Sauli.

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
