## [Decision Letter · Decision Letter 0]

30 Apr 2021

PONE-D-21-05111

Expressions of p16 and TOP2A protein biomarkers in cervical cancer lesions and their correlation with clinico-histopathological characteristics in Tanzania

PLOS ONE

Dear Dr. Zuberi,

Thank you for submitting your manuscript to PLOS ONE. After careful consideration, we feel that it has merit but does not meet PLOS ONE’s publication criteria as it currently stands. Therefore, we invite you to submit a revised version of the manuscript that addresses the points raised during the review process.

It is critical to address the methodological issues as well as the data presentation in the manuscript.

Also, the reviewers have indicated that important references are missing from both the introduction and discussion. Discussion should further be improved by expanding some topics especially the possible use of these assays in Tanzania.

We look forward to receiving your revised manuscript.

Kind regards,

Ivan Sabol

Academic Editor

PLOS ONE

Additional Editor Comments:

Abstract. Age can be rounded to 1 decimal point like was done for percentages

Sentence at P6L87-88 repeats the same idea from a previous paragraph P5L82-85. Can probably be condensed

Materials and methods P8 L137. Were any negative controls used/ tested?

Results P11 L199-200 sentence overlaps with the sentence at P11 L203-204. Can probably be condensed

If both p16 and TOP2a correlate with lesion grade and p16 correlates with TOP2a, it might be opportune to discuss which one works better(or at least good enough) or is their combination better (mandatory?). For example, TOP2a was negative in 2 CIN3 lesions and positive in some benign lesions?

Discussion Sentence at P11 L218-220 repeats the same idea from a previous paragraph P11 L215-217. Should be condensed to avoid repetition.

Sentence at P11L224-227 seems like a repeat from introduction. Doesnt expand on the information and lacks some reference.

Relatively low number of samples with precancerous lesions prevents drawing strong conclusions about the utility of those biomarkers in actual screening. However, since the biopsy material was collected in 2018, is it possible to obtain and present any follow-up data for women with precancerous lesions and whether those positive for tested biomarkers progressed further, and/or those found negative had lesions regress?

Table 1. Consider condensing the table so that „histological class“ totals are shown as subtotals for „Histopatological subtype“ categories instead of repeating the rows. Round the age numbers to 1 decimal place

Table 2. Instead of showing the „All (N=145)“ column percentages each with „(100)“, consider showing the percentages against the column total. In this way a part of information shown in Table 1 could be omitted (age and histopathological class).

What is the „simulated p value“ mentioned in Table 2. Fischers exact test is usually done on 2x2 tables. However, Table 2 doesnt have any 2x2 situations. Please elaborate on the statistical method/tool used to calculate Fischers exact test on larger tables. What is the critical difference between tests for „histopathological diagnosis“ and „histopathological subtype“? As the table is presented those p-values seem to be calculated in the same way? Possibly, it would be more informative to calculate the difference within each subtype?

The *** indicator is meaningless in Table 2 as actual p value is shown.

Table 3. How were the variables „Cancerous“ and „Benign“ coded? Why wasnt „Histopathological diagnosis“ with multiple levels used instead? Consider reordering the variables in a meaningful order (age shouldnt be shown between 2 different histological variables, but either as first or last variable)

Table 3 could be supplemented with the actual coefficients from Figure 3 and presented together in a table instead of both table and figure. Significance value indicators (*, ** and ***) are obsolete if actual p-values are shown in the table. However, consider showing only the relevant correlation coefficients and highlight those coefficients with matching * to *** indicator.

Data availability statement should be brought in line with PLOS data policy. Please add a large supplementary table showing the values behind the means, standard deviations and other measures reported. Ie the table should have a deidentified patient ID, age (and/or age group), diagnosis (and/or diagnosis group), clinical symptoms (and/or with grouping), p16 score, TOP2a score and show at least 145 rows

Journal Requirements:

Reviewers' comments:

Reviewer's Responses to Questions

**Comments to the Author**

1. Is the manuscript technically sound, and do the data support the conclusions?

Reviewer #1: Partly

Reviewer #2: Partly

2. Has the statistical analysis been performed appropriately and rigorously? 

Reviewer #1: No

Reviewer #2: I Don't Know

3. Have the authors made all data underlying the findings in their manuscript fully available?

Reviewer #1: Yes

Reviewer #2: Yes

4. Is the manuscript presented in an intelligible fashion and written in standard English?

Reviewer #1: Yes

Reviewer #2: Yes

5. Review Comments to the Author

Reviewer #1: In this manuscript, the authors analyze the utility of the p16 and TOP2A proteins as biomarkers for cervical cancer screening in patients from Tanzania. The study of p16 and TOP2A is not new, however, the study in this population makes it important. Therefore, the study is relevant, however the manuscript must be substantially improved before being considered for publication.

Major revisions

- The authors mention that 149 samples were included in the study, however only 139 samples are considered in the description of the diagnosis, line 182-184.

- Authors should improve the organization and description of results.

- In the results section, the authors describe the statistical differences found, however they fail to describe the biological relevance of their observations.

- The discussion should be improved, references are missing in some statements and the biological relevance of these proteins and their possible use as biomarkers in this study population should be deepened.

- The table 1 is irrelevant, similar information was included in table 2.

- In table 2, It is not clear to which comparison the p-value in the "Histopathological subtype" row refers.

- Statistical analysis is limited, the authors should at least delve into the risk involved in the increase in the expression of these proteins for the development of cancer, taking as reference benign lesions and precancerous lesions, for example using a logistic regression test using the age of the patients as a covariate to calculate ORs ajusted. This would provide much more relevant information to the study.

Minor revisions:

- In the abstract, the authors do not mention the type of sample that was analyzed or the relevant findings on the relationship between the expression of p16 and TOP2A proteins and the clinicopathological characteristics of the samples, being part of the title of the manuscript it would be important to be included.

- Considering that these proteins are already biomarkers reported in other parts of the world, the authors should focus the conclusion on the Tanzanian population and the usefulness of these markers in this population specifically.

- Review writing "Expression levels of p16 and have been reported..." line 76.

- "by rinsing with 20x diluted wash buffer", the authos must describe the composition of wash buffer, line 124

- "Thereafter, the sections were treated with counterstaining with hematoxylin..." counterstained??

- For clarity, the information included in Histopathological classification section (line 140) should be included in the "Retrieval of study samples and related clinical information" section, line 97.

- The authors mention "Four cervical biopsies were excluded due to either missing of tissue block or lack of clinical histopathological information" (line 179), If samples were excluded, should not be counted in the study population, simply mention that 145 biopsies were included.

Reviewer #2: While it would be of some interest, this study does not bring any breakthrough to the scientific community. Moreover, there are several major concerns.

The Introduction is missing the state-of-the art description of the cervical cancer prevention program in Tanzania, especially screening program. In addition, appropriate references related to Tanzania are missing (particularly on page 6, line 88-95), i.e.

- Mosha D, Mahande M, Ahaz J, Njau B, Kitali B, Obure J. Factors associated with management of cervical cancer patients at KCMC Hospital, Tanzania: a retrospective cross-sectional study. Tanzan J Health Res [Internet]. 2009;11(2).

- Urasa M, Darj E. Knowledge of cervical cancer and screening practices of nurses at a regional hospital in Tanzania. Afr Health Sci. 2011;11(1):48–57.

- Perng P, Perng W, Ngoma T, Kahesa C, Mwaiselage J, Merajver SD, et al. Promoters of and barriers to cervical cancer screening in a rural setting in Tanzania. Int J Gynaecol Obstet Off Organ Int Fed Gynaecol Obstet. 2013;123(3):221–5. doi: 10.1016/j.ijgo.2013.05.026.

- Lyimo FS, Beran TN. Demographic, knowledge, attitudinal, and accessibility factors associated with uptake of cervical cancer screening among women in a rural district of Tanzania: three public policy implications. BMC Public Health. 2012;12:22. doi: 10.1186/1471-2458-12-22.

- Dartell MA, Rasch V, Iftner T, Kahesa C, Mwaiselage JD, Junge J, et al. Performance of visual inspection with acetic acid and human papillomavirus testing for detection of high-grade cervical lesions in HIV positive and HIV negative Tanzanian women. Int J Cancer J Int Cancer. 2014;135(4):896–904. doi: 10.1002/ijc.28712.

- Qiao Weng, Jie Jiang, Fatma Mrisho Haji, Lamlet Hassan Nondo, Huaijun Zhou. Women’s knowledge of and attitudes toward cervical cancer and cervical cancer screening in Zanzibar, Tanzania: a cross-sectional study. BMC Cancer. 2020; 20: 63.

- etc.

The purpose of the study is not clear. I guess the authors wanted to improve the histopathological testing in their clinical hospital. The question is what did they achieve with this study. The conclusion being vague is not giving any feasible answer.

The Material and method and Results section are well presented in this paper. However, the discussion is repeating the results that is unnecessary. The sensitivity and specificity of histopathological biomarkers should be evaluated and discussed systematically.

Minor corrections related to text editing:

- Page 10, Line 194: delete 194 respectively

- Page 10, Line 198 and Page 13, Line 262: replace Additionally by In additin

- Page 11, Line 213: replace 1-year by one-year

- Page 11, Line 217: finish the sentence with a point

My recommendation is a major revision of the paper with the emphasis on the state-of-the art description of the cervical cancer prevention program and the histopathology as a diagnostic and not screening tool in cervical cancer prevention. The purpose of the study (Introduction) and the benefit of the study (Conclusion) should be clearly stated, while the Discussion completely rewritten.

6. PLOS authors have the option to publish the peer review history of their article (what does this mean?). If published, this will include your full peer review and any attached files.

Reviewer #1: No

Reviewer #2: No

---

## [Author Response · Author response to Decision Letter 0]

14 Jun 2021

Response to reviewer’s comments: 

Ref: PONE-D-21-05111

Title: Expressions of p16 and TOP2A protein biomarkers in cervical cancer lesions and their correlation with clinico-histopathological characteristics in Tanzania

Journal: PLOS ONE

Dear Editor,

Thank you very much for giving us an opportunity to revise our manuscript. We very much appreciate your reviewers for the positive and constructive comments on our manuscript. Based on comments we received, careful modifications have been made to the manuscript as will be seen in the re submitted version of this manuscript. All changes are in track changes mode. We hope the revised manuscript will meet the standards for the PLoS ONE journal. Please, also find our point-by-point responses to the raised queries/comments. The revised manuscript has been resubmitted. Please address any correspondence regarding the manuscript to me directly. We look forward to your positive response.

Thank you for your kind considerations!

Sincerely,

Zavuga Zuberi and authors

Additional Editor Comments

Abstract. Age can be rounded to 1 decimal point like was done for percentages

Response: Thank you for your comment. Age has been rounded to 1 decimal point throughout the manuscript as recommended.

Sentence at P6L87-88 repeats the same idea from a previous paragraph P5L82-85. Can probably be condensed

Response: Thank you for your good observation. The sentence at P6L87-88 has now been deleted.

Materials and methods P8 L137. Were any negative controls used/ tested?

Response: Thank you for your comment. In reducing false-positive IHC results, only two positive controls were used; for p16, a cervical squamous cell carcinoma tissue was used while for TOP2A, a breast carcinoma tissue was used (P8L157-159). However, no any negative controls were used. This information has been updated in the manuscript as will be seen.

Results P11 L199-200 sentence overlaps with the sentence at P11 L203-204. Can probably be condensed

Response: Thank you for your good observation. The sentence at P11L199-200 has now been deleted.

If both p16 and TOP2a correlate with lesion grade and p16 correlates with TOP2a, it might be opportune to discuss which one works better(or at least good enough) or is their combination better (mandatory?). For example, TOP2a was negative in 2 CIN3 lesions and positive in some benign lesions?

Response: Thank you for your good comment. About 40% difference in the expression of p16 (high) versus TOP2A (high) biomarkers was observed in 100% versus 60% of squamous cell carcinoma, which may imply that p16 biomarker is more specific (hence works better) for squamous cell carcinoma than TOP2A biomarker (P19L316-318). Nearly 17% decreased sensitivity of TOP2A relative to p16 was observed in discriminating cancerous from non-cancerous cervical lesions. This implies that, 20% diseased women (false-negatives) will be misdiagnosed. Therefore, p16 biomarker was found to be more specific and sensitive in accurately differentiating cancerous and non-cancerous cervical lesions than TOP2A among the studied women (P20L337-343).

Discussion Sentence at P11 L218-220 repeats the same idea from a previous paragraph P11 L215-217. Should be condensed to avoid repetition.

Response: Thank you for your good observation. The sentence at P11L218-220 has been deleted.

Sentence at P11L224-227 seems like a repeat from introduction. Doesnt expand on the information and lacks some reference.

Response: Thank you for your good observation. The sentence at P11L224-227 has been deleted.

Relatively low number of samples with precancerous lesions prevents drawing strong conclusions about the utility of those biomarkers in actual screening. However, since the biopsy material was collected in 2018, is it possible to obtain and present any follow-up data for women with precancerous lesions and whether those positive for tested biomarkers progressed further, and/or those found negative had lesions regress?

Response: Thank you for your comment. We agree that relatively low cases of precancerous lesions hinder us from drawing a strong conclusion about diagnostic utility of p16/TOP2A biomarkers for cervical cancer screening. Furthermore, it could be very useful to conduct a bigger follow-up study for assessing the progression or regression of the precancerous lesions. Importantly, this has been positively taken, and we hope to accommodate this in a guaranteed future study that will prospectively follow the study participants in a case-control study, aiming at understanding the expression pattern of these biomarkers (from mRNA level) in relation to the progression status of cervical lesions in the study area. Furthermore, these markers will be associated with HPV genotypes from the study area.

Table 1. Consider condensing the table so that „histological class“ totals are shown as subtotals for „Histopatological subtype“ categories instead of repeating the rows. Round the age numbers to 1 decimal place

Response: Thank you for your very useful comment. All the totals for histological class are shown as subtotals for histological subtype categories. Also, the age numbers have been round 1 decimal point.

Table 2. Instead of showing the „All (N=145)“ column percentages each with „(100)“, consider showing the percentages against the column total. In this way a part of information shown in Table 1 could be omitted (age and histopathological class).

Response: Thank you for your very useful comment. Percentages are now shown against the column total as advised. Part of information for age and histopathological class in Table 1 are omitted.

What is the „simulated p value“ mentioned in Table 2. Fischers exact test is usually done on 2x2 tables. However, Table 2 doesnt have any 2x2 situations. Please elaborate on the statistical method/tool used to calculate Fischers exact test on larger tables. 

response: Thank you for your comment. Yes, table 2 doesn’t obey the 2x2 contingency table, therefore an argument, stimulate.p.value was set equal to TRUE in the function fisher.test in R – this logical argument when set to TRUE computes p-values by Monte Carlo simulation, in larger than 2 by 2 tables.

What is the critical difference between tests for „histopathological diagnosis“ and „histopathological subtype“? As the table is presented those p-values seem to be calculated in the same way? Possibly, it would be more informative to calculate the difference within each subtype?

The *** indicator is meaningless in Table 2 as actual p value is shown.

Response: Thank you for your very useful comment. The statistical difference using fisher exact test for the histopathological subtype has been revised to calculate the difference within each subtype i.e., benign, precancerous and cancerous cervical lesions. The *** indicator in Table 2 has been deleted as advised.

Table 3. How were the variables „Cancerous“ and „Benign“ coded? Why wasnt „Histopathological diagnosis“ with multiple levels used instead? Consider reordering the variables in a meaningful order (age shouldnt be shown between 2 different histological variables, but either as first or last variable)

Response: Thank you for your comment. The coding for histopathological class relied on the type of cervical lesion observed. For example, if one of the diagnosed cervical lesions was ‘benign’ it was coded ‘1’ and other respective classes i.e., precancerous and cancerous lesions were all coded ‘0’. This coding was done for all the 145 cervical biopsied samples except for all non-definitive diagnosis samples were all coded ‘0’. Moreover, correlation variables in Table 3 have been ordered in as follows: age, benign, precancerous, cancerous, TOP2A and p16.

Table 3 could be supplemented with the actual coefficients from Figure 3 and presented together in a table instead of both table and figure. Significance value indicators (*, ** and ***) are obsolete if actual p-values are shown in the table. However, consider showing only the relevant correlation coefficients and highlight those coefficients with matching * to *** indicator.

Response: Thank you for your comment. Table 3 has been supplemented with actual correlation coefficients with significance levels highlighted using * to *** matching indicator.

Data availability statement should be brought in line with PLOS data policy. Please add a large supplementary table showing the values behind the means, standard deviations and other measures reported. Ie the table should have a deidentified patient ID, age (and/or age group), diagnosis (and/or diagnosis group), clinical symptoms (and/or with grouping), p16 score, TOP2a score and show at least 145 rows

Response: Thank you for your comment. S1 Table. Deidentified raw dataset for the study participants (n=149) seeking cervical cancer care were retrieved in a one-year period between May 1, 2017 and May 10, 2018 has been included.

Journal Requirements:

Response: Thank you for your comment. The manuscript has been formatted as per the PLoS One’s style requirements. We believe the current format now meets the publication standards for PLoS One.

https://journals.plos.org/plosone/s/file?id=wjVg/PLOSOne_formatting_sample_main_body.pdfand

 

Reviewer #1

Reviewer #1: In this manuscript, the authors analyze the utility of the p16 and TOP2A proteins as biomarkers for cervical cancer screening in patients from Tanzania. The study of p16 and TOP2A is not new, however, the study in this population makes it important. Therefore, the study is relevant, however the manuscript must be substantially improved before being considered for publication.

Response: Thank you for your comment. However, substantial changes have been made to improve the manuscript as will be reviewed.

Major revisions

- The authors mention that 149 samples were included in the study, however only 139 samples are considered in the description of the diagnosis, line 182-184.

Response: Thank you for your comment. This study was a retrospective study that retrieved 149 FFPE in a one-year period from May 1, 2017 to May 10, 2018. About 145 FFPE samples were included for immunohistochemistry staining for p16 and TOP2A biomarkers. The remaining 4 FFPE samples were excluded due to either missing of tissue block or lack of clinical histopathological information. In addition, the 6 FFPE samples were histologically classified as ‘non-definitive diagnosis’, and during IHC analysis they expressed low expression levels for both p16 and TOP2A biomarkers.

- Authors should improve the organization and description of results.

Response: Thank you for your comment. The organization of results have been substantially improved. Specifically, part of information such as age categories and histological classes have omitted/ merged in the Table 1. Moreover, all the ‘100’ in the column with ‘N=145’ has been modified to include percentages in Table 2. In addition, Figure 3 has been deleted and the correlation coefficients with significance levels highlighted using * to *** matching indicator are presented in Table 3.

Description of results 

- In the results section, the authors describe the statistical differences found, however they fail to describe the biological relevance of their observations.

Response: Thank you for your good comment. The biological relevance of different observations has been revised and reported. For instance, 

(1) “These biomarkers may be potentially used for screening cervical cancer lesions” (P13L231-232). 

(2) “This implies that, p16 and TOP2A cannot be potentially used as prognostic biomarkers across all age groups” (P13L234-235). 

(3) “Therefore, cancerous cervical lesions strongly correlated with p16 and TOP2A expressions among the women studied. Also, p16 expression moderately correlated with TOP2A expression” (P15L249-251); and 

(4) “These findings imply that p16 may be a potential biomarker in discriminating cancerous cervical lesions from benign and precancerous cervical lesions among the women studied” (P16L265-267).

- The discussion should be improved, references are missing in some statements and the biological relevance of these proteins and their possible use as biomarkers in this study population should be deepened.

Response: Thank you for very useful comment. The discussion part has been thoroughly revised and the biological relevance of p16/TOP2A with their possible use in the population being included. For instance, “p16, a tumour suppressor protein that accumulates in the nucleus and cytoplasm of cervical cells transformed by high-risk HPV [22, 26, 28]. In this study, the sensitivity and specificity of p16 were greater than 90% in discriminating cancerous cervical lesions from non-cancerous cervical lesions, suggesting it to be very useful marker in the diagnosis between healthy and diseased women seeking cervical cancer care at KCMC which is in agreement with findings reported by Espinosa et al. [37]. Similarly, TOP2A, a nuclear protein that controls DNA replication and chromosome separation of aberrant cervical cells in S-phase of cell cycle [20, 22]. Nevertheless, a 17% decreased sensitivity of TOP2A relative to p16 was observed. This implies that, 20% diseased women (false-negatives) will be misdiagnosed. Our findings are in agreement with the sensitivity and specificity of mRNA TOP2A biomarker as reported in previous studies [38, 39]. Generally, p16 demonstrated to confer highly accurate diagnostic performance than TOP2A biomarker in effectively differentiating cancerous and non-cancerous cervical lesions among the studied women” (P20L333-345).

- The table 1 is irrelevant, similar information was included in table 2.

Response: Thank your comment. Part of information such as age categories and histological classes have omitted/ merged in the Table 1. Moreover, all the ‘100’ in the column with ‘All (N=145)’ has been modified to include percentages in Table 2.

- In table 2, It is not clear to which comparison the p-value in the "Histopathological subtype" row refers.

Response: Thank you for your very useful comment. The statistical difference using fisher exact test for the histopathological subtype has been revised to calculate the difference within each subtype i.e., benign, precancerous and cancerous cervical lesions. Different p-values within each subtype for p16 and TOP2A biomarkers have been reported. Kindly see the revised Table 2. 

- Statistical analysis is limited, the authors should at least delve into the risk involved in the increase in the expression of these proteins for the development of cancer, taking as reference benign lesions and precancerous lesions, for example using a logistic regression test using the age of the patients as a covariate to calculate ORs ajusted. This would provide much more relevant information to the study.

Response: Thank you for your very useful comment. The statistical analyses have been extended by: (1) assessing the predictive risk of p16/TOP2A biomarkers with development of cervical cancer lesions in a logistic regression model; and (2) the diagnostic utility of p16/TOP2A biomarkers was evaluated by assessing the sensitivity, specificity and accuracy in discriminating cancerous from non-cancerous cervical lesions (P9L176-181, Table 4 and 5).

Minor revisions:

- In the abstract, the authors do not mention the type of sample that was analyzed or the relevant findings on the relationship between the expression of p16 and TOP2A proteins and the clinicopathological characteristics of the samples, being part of the title of the manuscript it would be important to be included.

Response: Thank you for your good comment. The type of sample analyzed has been mentioned in the abstract (P2L34-36). The results and conclusion parts have been re-written as per the editor’s and reviewers’ and comments (P2L38-52).

- Considering that these proteins are already biomarkers reported in other parts of the world, the authors should focus the conclusion on the Tanzanian population and the usefulness of these markers in this population specifically.

Response: Thank you for your good comment. The conclusion has been revised and now reads “p16 and TOP2A proteins may be promising biomarkers for diagnosis of cancerous cervical lesions from non-cancerous cervical lesions among women seeking cervical cancer care in Tanzania. However, further investigation and cost-benefits analyses are still needed before considering these proteins as biomarkers for diagnosis of cervical cancer in Tanzania and other low-resources countries with high-burden of cervical cancer cases. In addition, specificity and sensitivity performances of these biomarkers need to be validated and compared to readily available commercial diagnostic assays for cervical cancer. Importantly, a large study involving a combination of p16 and TOP2A biomarkers and their association with high-risk HPV subtypes can provide a more detailed pattern in the study area” (P20L346-354).

- Review writing "Expression levels of p16 and have been reported..." line 76.

Response: Thank you for your comment. The sentence has been rewritten to “Expression levels of p16 and TOP2A biomarkers have been reported in the screening and diagnosis of cervical cancer in high-resource countries” (P5L97-98).

- "by rinsing with 20x diluted wash buffer", the authos must describe the composition of wash buffer, line 124

Response: Thank you for your comment. The EnVision™ FLEX Mini Kit, High pH (Dako Denmark A/S, Glostrup, Denmark) that has EnVision™ FLEX Wash Buffer (20x) was used for carrying out IHC analyses for p16 and TOP2A biomarkers. 

- "Thereafter, the sections were treated with counterstaining with hematoxylin..." counterstained??

Response: Thank you for your very good comment. The sentence has been rewritten to “Thereafter, the sections were counterstained with hematoxylin for 17 dips and bluing for 5 min” (P8L154).

- For clarity, the information included in Histopathological classification section (line 140) should be included in the "Retrieval of study samples and related clinical information" section, line 97.

Response: Thank you for your comment. The information that was included in the subsection ‘histopathological classification’ are included in the subsection of “Retrieval of study samples and related clinical information” (P7L124-130). The subsection titled ‘Histopathological classification’ has been deleted.

- The authors mention "Four cervical biopsies were excluded due to either missing of tissue block or lack of clinical histopathological information" (line 179), If samples were excluded, should not be counted in the study population, simply mention that 145 biopsies were included.

Response: Thank you for your good comment. This paragraph has been rewritten to “A total of 145 cervical biopsies and patient’s information retrieved from the Pathology department archival material between May 1, 2017 and May 10, 2018 were included in this study. All the cervical biopsies were used for the immunohistochemistry staining of p16 and TOP2A biomarkers” (P10L193-195).

Reviewer #2

Reviewer #2: While it would be of some interest, this study does not bring any breakthrough to the scientific community. Moreover, there are several major concerns.

Response: Thank you for your comment. We believe that all the raised concerns have been fully addressed to meet the requirements and publication standards of PLoS One journal.

The Introduction is missing the state-of-the art description of the cervical cancer prevention program in Tanzania, especially screening program. In addition, appropriate references related to Tanzania are missing (particularly on page 6, line 88-95), i.e.

- Mosha D, Mahande M, Ahaz J, Njau B, Kitali B, Obure J. Factors associated with management of cervical cancer patients at KCMC Hospital, Tanzania: a retrospective cross-sectional study. Tanzan J Health Res [Internet]. 2009;11(2).

- Urasa M, Darj E. Knowledge of cervical cancer and screening practices of nurses at a regional hospital in Tanzania. Afr Health Sci. 2011;11(1):48–57.

- Perng P, Perng W, Ngoma T, Kahesa C, Mwaiselage J, Merajver SD, et al. Promoters of and barriers to cervical cancer screening in a rural setting in Tanzania. Int J Gynaecol Obstet Off Organ Int Fed Gynaecol Obstet. 2013;123(3):221–5. doi: 10.1016/j.ijgo.2013.05.026.

- Lyimo FS, Beran TN. Demographic, knowledge, attitudinal, and accessibility factors associated with uptake of cervical cancer screening among women in a rural district of Tanzania: three public policy implications. BMC Public Health. 2012;12:22. doi: 10.1186/1471-2458-12-22.

- Dartell MA, Rasch V, Iftner T, Kahesa C, Mwaiselage JD, Junge J, et al. Performance of visual inspection with acetic acid and human papillomavirus testing for detection of high-grade cervical lesions in HIV positive and HIV negative Tanzanian women. Int J Cancer J Int Cancer. 2014;135(4):896–904. doi: 10.1002/ijc.28712.

- Qiao Weng, Jie Jiang, Fatma Mrisho Haji, Lamlet Hassan Nondo, Huaijun Zhou. Women’s knowledge of and attitudes toward cervical cancer and cervical cancer screening in Zanzibar, Tanzania: a cross-sectional study. BMC Cancer. 2020; 20: 63.

- etc.

Response: Thank you for your good comment. The introduction part has been revised to include information about cervical cancer screening program in Tanzania. Appropriate references from Perng et al., 2013; Lyimo et al., 2012; Dartell et al., 2014; and Weng et al., 2020 were used to describe the state-of-the-art cervical cancer screening program in Tanzania (P4L73-76, P5L81-83).

The purpose of the study is not clear. I guess the authors wanted to improve the histopathological testing in their clinical hospital. The question is what did they achieve with this study. The conclusion being vague is not giving any feasible answer.

Response: Thank you for your good comment. The purpose of the study “… aimed at investigating the expression of p16 and TOP2A proteins as diagnostic markers for cervical cancer lesions in correlation with the clinico-histopathological features among women seeking cervical cancer care at Kilimanjaro Christian Medical Centre, Tanzania” (P6L106-109).

The Material and method and Results section are well presented in this paper. However, the discussion is repeating the results that is unnecessary. The sensitivity and specificity of histopathological biomarkers should be evaluated and discussed systematically.

Response: Thank you for your good comment. Diagnostic values i.e., sensitivity, specificity and accuracy of p16/TOP2A biomarkers were evaluated and systematically discussed in discriminating cancerous from non-cancerous cervical lesions (P16L258-267, Table 4). 

For instance, “p16, a tumour suppressor protein that accumulates in the nucleus and cytoplasm of cervical cells transformed by high-risk HPV [22, 26, 28]. In this study, the sensitivity and specificity of p16 were greater than 90% in discriminating cancerous cervical lesions from non-cancerous cervical lesions, suggesting it to be very useful marker in the diagnosis between healthy and diseased women seeking cervical cancer care at KCMC which is in agreement with findings reported by Espinosa et al. [37]. Similarly, TOP2A, a nuclear protein that controls DNA replication and chromosome separation of aberrant cervical cells in S-phase of cell cycle [20, 22]. Nevertheless, a 17% decreased sensitivity of TOP2A relative to p16 was observed. This implies that, 20% diseased women (false-negatives) will be misdiagnosed. Our findings are in agreement with the sensitivity and specificity of TOP2A mRNA biomarker as reported in previous studies [38, 39]. Generally, p16 demonstrated to confer highly accurate diagnostic performance than TOP2A biomarker in effectively differentiating cancerous and non-cancerous cervical lesions among the studied women” (P20L333-345).

Minor corrections related to text editing:

- Page 10, Line 194: delete 194 respectively

Response: Thank you for your good comment. The word ‘respectively’ has been deleted.

- Page 10, Line 198 and Page 13, Line 262: replace Additionally by In addition

Response: Thank you for good comment. The word ‘Additionally’ has been replaced by the word ‘In addition’ throughout the whole manuscript.

- Page 11, Line 213: replace 1-year by one-year

Response: Thank you for your good comment. The word ‘1-year’ has been replaced by the word ‘one-year’.

- Page 11, Line 217: finish the sentence with a point

Response: Thank you for your good comment. The sentence has been modified to read as follows ‘These study findings correspond with previously reported findings that suggested p16 and TOP2A immune expression increases with severity of cervical lesions, and may thus be used as markers in the classification of cervical lesions’.

My recommendation is a major revision of the paper with the emphasis on the state-of-the art description of the cervical cancer prevention program and the histopathology as a diagnostic and not screening tool in cervical cancer prevention. The purpose of the study (Introduction) and the benefit of the study (Conclusion) should be clearly stated, while the Discussion completely rewritten.

Response: Thank you for very useful comments. The significant improvements have been done in improving the introduction, discussion, and conclusion parts. We believe that our revised manuscript will meet the publication standards of PLoS One.

---

## [Decision Letter · Decision Letter 1]

5 Jul 2021

PONE-D-21-05111R1

Expressions of p16 and TOP2A protein biomarkers in cervical cancer lesions and their correlation with clinico-histopathological characteristics in Tanzania

PLOS ONE

Dear Dr. Zuberi,

Thank you for submitting your manuscript to PLOS ONE. After careful consideration, we feel that the revised manuscript has merit but does not fully meet PLOS ONE’s publication criteria as it currently stands. Therefore, we invite you to submit a revised version of the manuscript that addresses the points raised during the review process.

While the manuscript was significantly improved, one critical issue remains. Namely, the authors use biopsy material throughout the study. Biopsy material should not be considered a part of the cervical screening process. On the other hand, the authors give a lot of emphasis on the usability of the tested markers in the screening despite not testing them in the screening setting (i.e. lines 52, 64-65, 106, 109, 341, 352). The manuscript should be revised so that it is evident that results obtained on cervical biopsies are not in fact intended for screening purposes and that cervical cancer screening is unlikely to ever include imunohistochemical marker staining of cervical biopsies as routine. Referring everyone for biopsy is not a good screening strategy. Also, the authors imply their study goal was to address the lack of „low-cost, sensitive and specific screening methods... in Tanzania. Therefore…“ (line 106). The data obtained on (mostly) cervical cancer biopsies only implies that those markers might also work well in a screening setting but the current manuscript cannot provide solid evidence to that end.

On the other hand, the authors fail to significantly explore the utility of having p16/TOP2A results on the diagnosis and/or prognosis of cervical cancer in Tanzania despite this being mentioned (lines 101, 105, 109). For example, would having those results clarify the „Non-definitive diagnosis” cases, how common are “non-definitive diagnosis” cases overall and would additional biomarker results affect treatment and possibly prognosis/survival of those or all patients?

Overall, while the methods are clearly described and the experiments conducted adequately, the conclusions and the interpretation of the data is not appropriate (Publication criteria 4) and needs to be revised, expanded or at least rephrased extensively. Screening should be de-emphasized throughout the manuscript and only briefly mentioned as a possible use of those markers since the presented data doesnt address this issue.

We look forward to receiving your revised manuscript.

Kind regards,

Ivan Sabol

Academic Editor

PLOS ONE

Additional Editor Comments (if provided):

Typo: L80 „frequentky”

L159 How does

Software error L202 „(Table 1Error! Reference source not found.).”

Software error L208, L212, 214, 216,249 as above

L275 sentence unclear „Nevertheless, cancerous cervical lesions were significantly regressed with the expression of p16 than non-cancerous cervical lesions“

Typo L338 "p1"

L258, instead of coding Benign lesions as 1 and all other lesions as 0 for the Table 3 correlations, it might be better to simply code disease stages as 0 for benign, 1 for precancer and 2 as cancer and calculate the correlation with this one parameter instead of 3 different binary parameters for a single variable (histopathological class).

Reviewers' comments:

Reviewer's Responses to Questions

**Comments to the Author**

1. If the authors have adequately addressed your comments raised in a previous round of review and you feel that this manuscript is now acceptable for publication, you may indicate that here to bypass the “Comments to the Author” section, enter your conflict of interest statement in the “Confidential to Editor” section, and submit your "Accept" recommendation.

Reviewer #2: (No Response)

Reviewer #3: All comments have been addressed

2. Is the manuscript technically sound, and do the data support the conclusions?

Reviewer #2: Partly

Reviewer #3: Yes

3. Has the statistical analysis been performed appropriately and rigorously? 

Reviewer #2: I Don't Know

Reviewer #3: No

4. Have the authors made all data underlying the findings in their manuscript fully available?

Reviewer #2: Yes

Reviewer #3: Yes

5. Is the manuscript presented in an intelligible fashion and written in standard English?

Reviewer #2: Yes

Reviewer #3: Yes

6. Review Comments to the Author

Reviewer #2: Although the manuscript was somehow improved the authors did not address all my comments raised in a previous round of review.

Major deficiency of the study to resolve

As mentioned in my previous review, the clear goal of the study would be desirable taking into consideration the basic knowledge on CC screening and diagnostic procedures. The study needs to clearly distinguish the screening test from confirmation testing, that is the case in this study on archival biopsies. HPV test results predict the risk of cervical cancer and its precursors (cervical intraepithelial neoplasia) better than cytological or colposcopic/visual abnormalities, which indicate only signs of HPV infection.

The results of HPV testing would add value to the study as the limitation of the study is a relatively low number of analysed samples (n=145). Therefore, the authors are advised to provide those results.

It would be interesting to refer to the crude and effective cervical cancer screening coverage for women aged 25 to 64 years in Tanzania in the Introduction.

Major remarks

L80-81 delete “These methods have poor sensitivity and specificity and are very subjective.” This statement is inappropriate. In L77 the authors are saying that “there are inadequate immunohistopathological testing facilities”, which explains why the most common screening CC test (Pap test) is not implemented in Tanzania. The authors can say, of cause, if it is the case “These methods require well-educated and qualified staff as well as high quality control in test implementation, that, unfortunately is not provided in Tanzania”.

L81-83 “According to Dartell et al. [11] reported lower sensitivity of visual inspection with acetic acid than HPV-testing in detecting HSIL cervical lesions during a hospital-based cervical cancer screening event in Dar es Salaam, Tanzania.” Indeed, HPV-testing as primary screening is considered a gold standard in developing countries, especially because of the lack of good cytology facilities. My advice to authors is to check the worldwide literature on CC screening programs, especially those published and acknowledged by IARC and WHO. Thus, complete the paragraph with the reasoning leading to the adequate use of cellular biomarkers in this study.

L86, L98, L100 These immunohistological test are not appropriate for screening, although useful to improve diagnostic of CC on biopsy were cytology fails. Thus, reconsider this phrase and the whole purpose of your study.

L106 Pay attention to the use of cheap HPV tests design for the developing countries rather that immunohistological test, which are not cheap at all.

L109-L110. I am not convinced of the screening use of the mentioned biomarkers. So, reconsider the purpose of your study.

Minor corrections to do

L80 delete “Pap smear test” and replace with “conventional cytology (Pap test)”

L85, L95 replace p16 with its full name, eventually in bracket p16.

L109: replace “save” by “serve”

L254: .... and TOP2A expressions among (Table 2). Finish the sentence.

L255 replace “Also,” with “In addition,”

Lines 244, 251 and elsewhere: "among the studied women" replace by "in the studied population"

L249 ? Error! Reference source not found.

L250 delete “Nevertheless,”

L285 “Overall, our findings suggest that p16 and TOP2A biomarkers could be associated with the development of cervical cancer lesions.” This statement is irrelevant as both biomarkers are commercially available as cervical cancer biomarkers. So, please rephrase your conclusion.

Replace everywhere “100.0% and 0.0%” by “100% and 0%”

Table 2 delete a double bracket: 1 (14.3))

Reviewer #3: The authors have notably improved the manuscript compared to the first version, considering the comments of the reviewers. Although the study of these proteins as biomarkers in cervical cancer is not new, the data are important due to their application in the study population. The article could be published in its current form.

Minor revisions

In Table 5, the authors must mention which variable they used to adjust the ORs; Were the ORs adjusted for age?

7. PLOS authors have the option to publish the peer review history of their article (what does this mean?). If published, this will include your full peer review and any attached files.

Reviewer #2: No

Reviewer #3: No

---

## [Author Response · Author response to Decision Letter 1]

1 Sep 2021

Editors’ Comments

While the manuscript was significantly improved, one critical issue remains. Namely, the authors use biopsy material throughout the study. Biopsy material should not be considered a part of the cervical screening process. On the other hand, the authors give a lot of emphasis on the usability of the tested markers in the screening despite not testing them in the screening setting (i.e. lines 52, 64-65, 106, 109, 341, 352). The manuscript should be revised so that it is evident that results obtained on cervical biopsies are not in fact intended for screening purposes and that cervical cancer screening is unlikely to ever include imunohistochemical marker staining of cervical biopsies as routine. Referring everyone for biopsy is not a good screening strategy. Also, the authors imply their study goal was to address the lack of „low-cost, sensitive and specific screening methods... in Tanzania. Therefore…“ (line 106). The data obtained on (mostly) cervical cancer biopsies only implies that those markers might also work well in a screening setting but the current manuscript cannot provide solid evidence to that end.

Response: Thank you for your comment. The usefulness of immunohistochemistry assay in improving diagnosis of cervical cancer lesions has been considered throughout the manuscript. For instance, P4L67-69, P5L84-85, P19L317, P20L349-351, etc.

On the other hand, the authors fail to significantly explore the utility of having p16/TOP2A results on the diagnosis and/or prognosis of cervical cancer in Tanzania despite this being mentioned (lines 101, 105, 109). For example, would having those results clarify the „Non-definitive diagnosis” cases, how common are “non-definitive diagnosis” cases overall and would additional biomarker results affect treatment and possibly prognosis/survival of those or all patients?

Response: Thank you for your comment. “Non-definitive diagnosis” was used to refer to the preliminary H&E results of the cervical biopsy. Based on the relatively low number (n=6) of non-definitive cases, they were excluded in the statistical analyses for the correlation, diagnostic performance and strength of associations as they may confer very minimal risk of cervical cancer development.

Overall, while the methods are clearly described and the experiments conducted adequately, the conclusions and the interpretation of the data is not appropriate (Publication criteria 4) and needs to be revised, expanded or at least rephrased extensively. Screening should be de-emphasized throughout the manuscript and only briefly mentioned as a possible use of those markers since the presented data doesnt address this issue.

Response: Thank you for your useful comments. Our conclusion and data interpretation have been re-written to de-emphasize about screening throughout the manuscript. The conclusion is now revised in a way that supports the data presented as highlighted in the publication criteria 4 (P22L374-380).

Additional Editor Comments

Typo: L80 „frequentky”

Response: Thank you for your comment. Typing error has been corrected (P5L85).

 L159 How does

Response: Thank you for your comment. Unfortunately, the concern raised is incomplete.

 Software error L202 „(Table 1Error! Reference source not found.).”

Response: Thank you for your comment. The error associated with document formatting has been corrected (P11L225).

Software error L208, L212, 214, 216,249 as above

Response: Thank you for your comment. All the software errors in the document have been corrected (P12L234, P12L236, P12L237, P16L272).

 L275 sentence unclear „Nevertheless, cancerous cervical lesions were significantly regressed with the expression of p16 than non-cancerous cervical lesions“

Response: Thank you for your good observation. The sentence has been revised to “Nevertheless, cancerous cervical lesions were statistically correlated with p16 expression” (P16L272).

 Typo L338 "p1"

Response: Thank you for your comment. Typing error has been corrected (P21L356).

L258, instead of coding Benign lesions as 1 and all other lesions as 0 for the Table 3 correlations, it might be better to simply code disease stages as 0 for benign, 1 for precancer and 2 as cancer and calculate the correlation with this one parameter instead of 3 different binary parameters for a single variable (histopathological class).

Response: Thank you for your good observation. The data for correlation test were recoded for disease stages as “0” for benign, “1” for precancerous and “2” for cancerous. Moreover, data inference was revised as per the calculated correlation output (Table 3 and P16L263-274). 

Reviewer #2

Reviewer #2: Although the manuscript was somehow improved the authors did not address all my comments raised in a previous round of review.

Response: Thank you for your comment. However, substantial changes have been made to improve the manuscript as will be reviewed.

Major deficiency of the study to resolve

As mentioned in my previous review, the clear goal of the study would be desirable taking into consideration the basic knowledge on CC screening and diagnostic procedures. The study needs to clearly distinguish the screening test from confirmation testing, that is the case in this study on archival biopsies. HPV test results predict the risk of cervical cancer and its precursors (cervical intraepithelial neoplasia) better than cytological or colposcopic/visual abnormalities, which indicate only signs of HPV infection.

Response: Thank you for your comment. Yes, we agree with the reviewer, this study basically was not intended to assess the screening tests for cervical cancer, rather we conceived the present study to evaluate the usefulness of p16 and TOP2A as potential biomarkers in dysplastic and malignant alteration of cervical epithelium by analyzing a series of benign, precancerous and cancerous cervical lesions so as to assess whether their expression might be of any use in predicting prognosis in cervical carcinogenesis (P6L130-134).

The results of HPV testing would add value to the study as the limitation of the study is a relatively low number of analysed samples (n=145). Therefore, the authors are advised to provide those results.

Response: Thank you for a very useful comment. However, due to limited financial support received as the part of Master’s studies, genotyping of HPV subtypes using archived cervical biopsies was not performed.

We kindly acknowledge the importance of genotyping HPV subtypes from the archived cervical biopsies in predicting the risks associated with the development of cervical cancer lesions and its precursors that may provide a different pattern in association with the expression levels of p16 and TOP2A biomarkers. This information has been updated in the manuscript (P21L369-373). This information has been taken care by recommending genotyping of HPV subtypes in a further study.

It would be interesting to refer to the crude and effective cervical cancer screening coverage for women aged 25 to 64 years in Tanzania in the Introduction.

Response: Thank you for your comment. “In Tanzania, very limited studies estimating the coverage of cervical cancer screening by age exist [12]. However, a follow-up study that enrolled women aged between 25 and 60 years from three cervical cancer screening clinics from urban and semi-rural areas in Tanzania reported 17.2% (696/4,043) and 14.2% (438/3,074) of the women aged between 25 and 60 years had high-risk HPV (hr-HPV) at baseline and in the first follow-up, respectively [13]. Only 3.4% (139/4,043) of the women aged between 25 and 60 years had high-grade squamous intraepithelial lesion (HSIL) in the same study [13]” (P5L93-99).

Major remarks

L80-81 delete “These methods have poor sensitivity and specificity and are very subjective.” This statement is inappropriate.

Response: Thank you for your good comment. The statement ‘These methods have poor sensitivity and specificity and are very subjective’ has been deleted’.

In L77 the authors are saying that “there are inadequate immunohistopathological testing facilities”, which explains why the most common screening CC test (Pap test) is not implemented in Tanzania. The authors can say, of cause, if it is the case “These methods require well-educated and qualified staff as well as high quality control in test implementation, that, unfortunately is not provided in Tanzania”.

Response: Thank you for your good comment. The sentence ‘there are inadequate immunohistopathological testing facilities’ has been deleted and replaced with the sentence you have advised (P5L81-83). However, Pap test is available in the zonal hospitals in Tanzania (P5L91-92).

L81-83 “According to Dartell et al. [11] reported lower sensitivity of visual inspection with acetic acid than HPV-testing in detecting HSIL cervical lesions during a hospital-based cervical cancer screening event in Dar es Salaam, Tanzania.” Indeed, HPV-testing as primary screening is considered a gold standard in developing countries, especially because of the lack of good cytology facilities. My advice to authors is to check the worldwide literature on CC screening programs, especially those published and acknowledged by IARC and WHO. Thus, complete the paragraph with the reasoning leading to the adequate use of cellular biomarkers in this study.

Response: Thank you for your good comment. The paragraph has been revised to “Another study by Dartell et al. [11] reported lower sensitivity of VIA than HPV-testing in detecting HSIL cervical lesions during a hospital-based cervical cancer screening event in Dar es Salaam, Tanzania. In many high-income countries, cervical cancer rates declined substantially after the wide spread introduction of the Pap test in the mid 20th century [12]. However, similar trends did not emerge in low- and middle-income countries including Tanzania due to lack of resources for screening implementation including cytology review and low population coverage leading to advanced detection of cervical cancer and poor survival rates [2, 12]” (P5L99-106).

L86, L98, L100 These immunohistological test are not appropriate for screening, although useful to improve diagnostic of CC on biopsy were cytology fails. Thus, reconsider this phrase and the whole purpose of your study.

Response: Thank you for your comment. The usefulness of immunohistochemistry assay in improving diagnosis of cervical cancer lesions has been considered throughout the manuscript. For instance, P6L109, P6L121, P6L123, etc.

L106 Pay attention to the use of cheap HPV tests design for the developing countries rather that immunohistological test, which are not cheap at all.

Response: Thank you for your very useful comment. The sentence has been revised to capture low-cost HPV DNA tests such as careHPV (P7L128-130).

L109-L110. I am not convinced of the screening use of the mentioned biomarkers. So, reconsider the purpose of your study.

Response: Thank you for your comment. This has been re-written throughout the manuscript and the markers herein were not studied for screening but rather as future possible diagnostic markers.

Minor corrections to do

L80 delete “Pap smear test” and replace with “conventional cytology (Pap test)”

Response: Thank you for your comment. The ‘Pap smear test’ has been replaced with ‘conventional cytology (Pap test)’ (P5L84).

L85, L95 replace p16 with its full name, eventually in bracket p16.

Response: Thank you for your comment. ‘Cyclin-dependent kinase inhibitor’ followed by p16 in bracket has been added (P2L33-34, P6L108). In addition, ‘cyclin-dependent kinase inhibitor’ has been deleted with p16 written without brackets.

L109: replace “save” by “serve”

Response: Thank you for your comment. The sentence that has the word ‘serve’ has been accommodated in the revised goal of study “…. usefulness of p16 and TOP2A as potential biomarkers in dysplastic and malignant alteration of cervical epithelium by analyzing a series of benign, precancerous and cancerous cervical lesions …...” (P7L130-134).

L254: .... and TOP2A expressions among (Table 2). Finish the sentence.

Response: Thank you for your comment. The sentence has been completed to include ‘in the studied population’ (P16L273-274).

L255 replace “Also,” with “In addition,”

Response: Thank you for your comment. The word ‘Also’ has been replaced with ‘In addition’.

Lines 244, 251 and elsewhere: "among the studied women" replace by "in the studied population"

Response: Thank you for your good comment. The words ‘among the studied women’ and/or ‘among studied women’ have been replaced with ‘in the studied population’ throughout the whole manuscript.

L249 ? Error! Reference source not found.

Response: Thank you for your comment. The error associated with document formatting has been corrected.

L250 delete “Nevertheless,”

Response: Thank you for your comment. The word ‘Nevertheless’ has been deleted.

L285 “Overall, our findings suggest that p16 and TOP2A biomarkers could be associated with the development of cervical cancer lesions.” This statement is irrelevant as both biomarkers are commercially available as cervical cancer biomarkers. So, please rephrase your conclusion.

Response: Thank you for your good comment. We have rephrased the sentence, “Overall, our findings suggest that p16 and TOP2A biomarkers could be associated with the development of cervical cancer lesions” to “Overall, our findings support the association of p16 and TOP2A biomarkers in the development of cervical cancer lesions” (P18L302-303).

Replace everywhere “100.0% and 0.0%” by “100% and 0%”

Response: Thank you for your comment. All values with ‘100.0%’ and ‘0.0%’ have been replaced by ‘100%’ and ‘0.0%’, respectively throughout the manuscript.

Table 2 delete a double bracket: 1 (14.3))

Response: Thank you for your comment. A second “closing” bracket has been deleted.

Reviewer #3

Reviewer #3: The authors have notably improved the manuscript compared to the first version, considering the comments of the reviewers. Although the study of these proteins as biomarkers in cervical cancer is not new, the data are important due to their application in the study population. The article could be published in its current form.

Response: Thank you for very constructive comments to improve our manuscript.

Minor revisions

In Table 5, the authors must mention which variable they used to adjust the ORs; Were the ORs adjusted for age?

Response: Thank you for good comment. The odd ratios were adjusted for age. Table 5 has been revised to show AOR which stands for ‘age-adjusted odds ratio’. In addition, the strength of associations between p16 and TOP2A biomarkers in differentiating cancerous cervical lesions from benign and precancerous cervical lesions is described in the context of age-adjusted odds ratio (P18L297-303).

---

## [Editor Report · Decision Letter 2]

13 Oct 2021

Expression analysis of p16 and TOP2A protein biomarkers in cervical cancer lesions and their correlation with clinico-histopathological characteristics in a referral hospital, Tanzania

PONE-D-21-05111R2

Dear Dr. Zuberi,

We’re pleased to inform you that your manuscript has been judged scientifically suitable for publication and will be formally accepted for publication once it meets all outstanding technical requirements.

Kind regards,

Ivan Sabol

Academic Editor

PLOS ONE

Additional Editor Comments (optional):

During the proof reading steps please delete additional "Error! Reference source not found." problems remaining in the text ie. P12 L236 and P16 L270
---

## [Editor Report · Acceptance letter]

18 Oct 2021

PONE-D-21-05111R2 

Expression analysis of p16 and TOP2A protein biomarkers in cervical cancer lesions and their correlation with clinico-histopathological characteristics in a referral hospital, Tanzania 

Dear Dr. Zuberi:

I'm pleased to inform you that your manuscript has been deemed suitable for publication in PLOS ONE. Congratulations! Your manuscript is now with our production department. 

Kind regards, 

on behalf of

Dr. Ivan Sabol 

Academic Editor

PLOS ONE